# Exosome-Transmitted tRF-16-K8J7K1B Promotes Tamoxifen Resistance by Reducing Drug-Induced Cell Apoptosis in Breast Cancer

**DOI:** 10.3390/cancers15030899

**Published:** 2023-01-31

**Authors:** Chunxiao Sun, Xiang Huang, Jun Li, Ziyi Fu, Yijia Hua, Tianyu Zeng, Yaozhou He, Ningjun Duan, Fan Yang, Yan Liang, Hao Wu, Wei Li, Yuchen Zhang, Yongmei Yin

**Affiliations:** 1Department of Oncology, The First Affiliated Hospital of Nanjing Medical University, Nanjing 210029, China; 2The First Clinical College, Nanjing Medical University, Nanjing 210029, China; 3Department of Oncology, Sir Run Run Hospital of Nanjing Medical University, Nanjing 211166, China; 4Department of Radiation Oncology, The Second Affiliated Hospital of Nanjing Medical University, Nanjing 210011, China; 5Jiangsu Key Lab of Cancer Biomarkers, Prevention and Treatment, Collaborative Innovation Center for Cancer Medicine, Nanjing Medical University, Nanjing 211166, China

**Keywords:** breast cancer, tamoxifen resistance, tRNA-derived fragments, exosomes

## Abstract

**Simple Summary:**

While the prognosis of hormone receptor-positive (HR+) breast cancer has been significantly improved, tamoxifen resistance remains a challenge in the treatment of HR+ breast cancer. This study identified that tRF-16-K8J7K1B, a novel small ncRNA derived from the 3′-end of tRNA^Ala-TGC^, was highly expressed in tamoxifen-resistant cells compared to parental cells. Moreover, extracellular tRF-16-K8J7K1B confers tamoxifen resistance via incorporation into exosomes and then degrades the expression of apoptosis-related proteins, reducing the proportion of drug-induced cell apoptosis. Therefore, we propose that exosomal tRF-16-K8J7K1B could be a potential predictive biomarker and therapeutic target for overcoming tamoxifen resistance.

**Abstract:**

Tamoxifen resistance remains a challenge in hormone receptor-positive (HR+) breast cancer. Recent evidence suggests that transfer ribonucleic acid (tRNA)-derived fragments play pivotal roles in the occurrence and development of various tumors. However, the relationship between tRNA-derived fragments and tamoxifen resistance remains unclear. In this study, we found that the expression of tRF-16-K8J7K1B was upregulated in tamoxifen-resistant cells in comparison with tamoxifen-sensitive cells. Higher levels of tRF-16-K8J7K1B were associated with shorter disease-free survival in HR+ breast cancer. Overexpression of tRF-16-K8J7K1B promotes tamoxifen resistance. Moreover, extracellular tRF-16-K8J7K1B could be packaged into exosomes and could disseminate tamoxifen resistance to recipient cells. Mechanistically, exosomal tRF-16-K8J7K1B downregulates the expression of apoptosis-related proteins, such as caspase 3 and poly (ADP-ribose) polymerase, by targeting tumor necrosis factor-related apoptosis-inducing ligand in receptor cells, thereby reducing drug-induced cell apoptosis. Therapeutically, the inhibition of exosomal tRF-16-K8J7K1B increases the sensitivity of breast cancer cells to tamoxifen in vivo. These data demonstrate that exosomal tRF-16-K8J7K1B may be a novel therapeutic target to overcome tamoxifen resistance in HR+ breast cancer.

## 1. Introduction

Breast cancer is the most common malignant tumor found in females [1]. Nearly 70% of patients with breast cancer are hormone receptor-positive (HR+), and endocrine therapy significantly improves the prognosis of HR+ breast cancer. As a selective estrogen receptor modulator, tamoxifen is widely used in endocrine therapy for patients with estrogen receptor (ER)-positive breast cancer [2]. However, nearly 30% of patients develop drug resistance during tamoxifen treatment [3], which poses significant challenges for subsequent regimens [4]. The mechanism of tamoxifen resistance is intricate, involving the interaction of multiple intracellular signaling pathways, which remain elusive. Several treatment options, including aromatase inhibitors, fulvestrant, and cyclin-dependent kinase 4 and 6 (CDK4/6) inhibitors, have been applied clinically to overcome tamoxifen resistance; however, their benefits are insignificant. Therefore, it is important to explore the molecular mechanisms of tamoxifen resistance and identify biomarkers to predict patients’ responses to tamoxifen.

Non-coding ribonucleic acids (ncRNAs) are a class of RNA that do not encode proteins [5]. tRNA derivatives are a novel class of small ncRNAs derived from precursors or mature tRNAs that are cleaved specifically by Dicer enzymes or angiogenin. Based on different restriction sites, tRNA derivatives can be divided into two subtypes: tRNA halves (tiRNAs) and tRNA-derived fragments (tRFs). tRFs are further categorized into four groups: tRF-1; tRF-3; tRF-5; and itRFs [6]. Previously, tRNA derivatives were considered non-functional fragments produced by random tRNA cleavage. However, an increasing amount of evidence indicates that tRFs have a regulatory effect on the biological processes of tumor proliferation, invasion, and drug resistance in human cancers [7,8]. Zhu et al. discovered that tRF-5026A inhibits gastric cancer cell proliferation by regulating the PTEN/PI3K/AKT signaling pathway [9]. tRF-LEU-CAG upregulated the expression of aurora kinase A in non-small-cell lung cancer, and knocking down tRF-LEU-CAG weakened its proliferative ability [10]. Additionally, tRFs are stably present in serum samples of human cancers and have been considered new biomarkers and therapeutic targets for cancer treatment [11,12]. Our previous study revealed that tRF-30-JZOyJE22RR33 and tRF-27-ZDXPHO53KSN are significantly upregulated in the serum of trastuzumab-resistant patients and may be potential biomarkers of trastuzumab resistance [13]. However, the role of tRFs in tamoxifen resistance has not been explored. Therefore, it is important to explore the molecular mechanisms of tRFs associated with tamoxifen resistance.

Exosomes are extracellular vesicles with diameters of 30–100 nm and are widely distributed in bodily fluids, such as serum, saliva, urine, and cerebrospinal fluid [14]. Exosomes can carry and promote the transfer of deoxyribonucleic acid, RNA, proteins, lipids, and other substances between cells; thus, they can regulate the occurrence and development of various diseases, including cancer. With the deepening of research, exosomes have been proven to play an important role in drug resistance [15]. Recent studies have revealed that exosomes derived from drug-resistant cells deliver long non-coding RNA small-nucleolar-RNA host gene 14 to sensitive cells, which can develop trastuzumab-resistant phenotypes by changing the expression of target genes and activating signaling pathways in breast cancer cells [16]. However, it remains unclear whether tRFs cause tamoxifen resistance in breast cancer and spread to recipient-sensitive cells by packaging into exosomes. In addition, inducing cell apoptosis is a key anticancer treatment strategy [17]. Tumor necrosis factor-related apoptosis-inducing ligand (TRAIL) could initiate the extrinsic apoptosis signaling pathway and are associated with tumor metastasis and resistance in multiple cancers [18]. Previous studies have indicated that triple-negative breast cancer is more susceptible to TRAIL than other subtypes of breast cancer cells [19]. At present, the relationship between TRAIL and drug resistance to endocrine therapy in HR+ breast cancer has not been fully clarified.

Here, we reveal the essential role of tRF-16-K8J7K1B in promoting tamoxifen resistance by its incorporation into exosomes. Moreover, we discovered that exosomal tRF-16-K8J7K1B downregulates the expression of apoptosis-related proteins by targeting TRAIL in receptor cells, thereby reducing the proportion of drug-induced cell apoptosis. Our data expound the molecular mechanism of exosomal tRF-16-K8J7K1B in promoting tamoxifen resistance and provide a new direction for treating tamoxifen-resistant patients.

## 2. Materials and Methods

### 2.1. Patient Samples

Fifty-six breast cancer serum samples were obtained from The First Affiliated Hospital of Nanjing Medical University after obtaining ethical approval (Approved No: 2011-SRFA-007) and informed consent from the patients. All patients were treated with tamoxifen during adjuvant therapy. We employed the unifying definition of tamoxifen primary resistance as follows: progression or new recurrences diagnosed within 24 months of adjuvant tamoxifen [20]. There were 27 patients who were tamoxifen-sensitive and 29 patients with tamoxifen primary resistance. The protocol for sample processing is detailed in our previous study [13]. All separated sera were stored at −80 °C until further processing. Detailed patient information is presented in Appendix A.

### 2.2. Cell Lines and Culture

The human ER-positive breast cancer cell lines MCF7 and T47D were obtained from the American Type Culture Collection. Tamoxifen-resistant cells (MCF7/TR and T47D/TR) were created by exposing their parent cells to a medium containing progressively increasing 4-hydroxytamoxifen (4-OHT) concentrations for more than 6 months. The obtained drug-resistant cells survived in the medium containing 20 µm/L 4-OHT concentration. All cell lines were cultured in Dulbecco’s modified Eagle’s medium supplemented with 10% fetal bovine serum (10099-141, GIBCO, CA, USA) and 1% penicillin or streptomycin (C100C5, NCM Biotech, Suzhou, China), and maintained in a 37 °C incubator with 5% CO_2_.

### 2.3. RNA Sequence Processing and Expression Analysis

Illumina NextSeq (Illumina, Inc., San Diego, CA, USA) 500 raw sequencing reads that passed the Illumina chastity filter were used for sequence analyses. After quality control, trimmed reads (with 5′,3′-adaptor based removed) were aligned to mature-tRNA and pre-tRNA reference sequences and filtered for more than 14 nt using Cutadapt software 1.17 (Sven Rahmann, Dortmund University of Technology, NRW, Dortmund, Germany). The sequencing reads were aligned with mature-tRNA on the entire genome using MINTbase v2.0 (https://cm.jefferson.edu/MINTbase, accessed on 5 August 2019). The sequencing data focused on aligning all tRNA derivatives. The tRF and tiRNA expression levels were measured and normalized to counts per million mapped reads. Statistical significance was set at *p* < 0.05.

### 2.4. Exosome Extraction, Identification, and Labeling

After cells were cultured in Dulbecco’s Modified Eagle’s Medium containing exosome-depleted fetal bovine serum (EX0-FBS-50A, SBI, USA) for 48 h, 150 mL cell medium was collected and centrifuged at 10,000× *g* for 30 min to remove dead cells and cell debris. The supernatant was filtered through a 0.22 µm filter (UFC910096-1, Millipore, Burlington, MA, USA) and ultracentrifuged at 120,000× *g* for 70 min. The above steps were repeated once, and the exosomes at the bottom were suspended in 50 µL phosphate-buffered saline (PBS). The concentration of exosomes was determined using a bicinchoninic acid protein kit. To identify the exosome morphology using transmission electron microscopy, 10 µL of fresh suspension was used, and the exosomes were fixed in 2% paraformaldehyde and adsorbed onto a copper net. For fluorescent labeling, PKH67 membrane dye (MINI67, Sigma, St. Louis, MO, USA) was added to the exosomes according to the manufacturer’s instructions. The green fluorescence of the labeled exosomes was observed using a laser confocal microscope after co-culturing with receptor cells. Image forming system: ×80k, HT Voltage: 120 kV, Beam Curr: 70 μA.

### 2.5. RNA Extraction and Quantitative Real-Time Polymerase Chain Reaction

Total RNA was extracted from the cells and exosomes using a TRIZOL reagent (15596026, Thermo Fisher Scientific, Inc., Waltham, MA, USA). For serum RNA isolation, all blood samples were operated following the protocols of the miRcute Serum or serum microRNA isolation kit (DP501, TIANGEN, Beijing, China). Complementary deoxyribonucleic acid was synthesized using the HiScript III RT SuperMix for quantitative polymerase chain reaction (PCR) (R233-01, Vazyme, Nanjing, China). Real-time PCR amplification was performed using the SYBR Green PCR Kit (RR820A, Takara, Kusatsu, Shiga, Japan), and the results were analyzed using the Light Cycler 480 II Real-Time PCR System (Roche, Basel, Switzerland). U6 and β-actin were used as internal references for detecting microRNA and messenger RNA (mRNA) expression levels. Primer sequences are listed in Appendix A.

### 2.6. Western Blotting

Cells and exosomes were extracted using a radioimmunoprecipitation assay buffer containing protease inhibitors. Protein samples (30 µg) were separated by sodium dodecyl sulfate-polyacrylamide gel electrophoresis and transferred to polyvinylidene fluoride membranes. After blocking with skimmed milk for 1 h, the polyvinylidene fluoride membrane was incubated with primary antibodies overnight, and then with the secondary antibody for 2 h. Finally, the proteins were visualized using an enhanced chemiluminescent system. All antibodies are listed in Appendix A.

### 2.7. Dual-Luciferase Reporter Assay

The 3′-untranslated (UTR) region fragments of TRAIL genes WT and MUT were inserted into the GP-miRGLO vector. WT or MUT luciferase plasmids were co-transfected with 5 × 10^4^ HEK293T cells, and tRF-16-K8J7K1B mimicked a negative control when the cells were fused to 70% density. After 24 h, 150 μL of passive lysis buffer was added to each well to harvest the cells, and 80 μL of the supernatant was used to determine luciferase activity using the dual-luciferase reporter assay system (11402ES60, Promega, Madison, WI, USA).

### 2.8. Animal Studies

Animal experiments were performed following the Institutional Animal Care and Use Committee guidelines, and the study was conducted at the Jiangsu Laboratory Animal Center (Approved No: 21080180). Five-week-old BALB/C nude mice were used to establish a breast cancer model. MCF-7 cells were harvested and mixed with 50% Matrigel (354234, BD Biosciences, Franklin Lakes, NJ, USA), and 1 × 10^7^ cells were injected into the mammary fat pads of female nude mice. After the tumor size reached 50 mm^3^, mice were randomly divided into four groups of ten each. The first group was injected intratumorally with 1 mL PBS every 3 d as a control, and the other three groups were injected intratumorally with 10 µg exosomes derived from MCF7, MCF7/Tamoxifen resistance-tRF-16-K8J7K1B negative control (MCF7/TR-nc), and MCF7/Tamoxifen resistance-tRF-16-K8J7K1B inhibitor (MCF7/TR-inhibitor) cells every 3 d. Each group was randomly divided into two groups of five each until the tumor volume reached 200 mm^3^. One group was administered 5 mg/kg tamoxifen (Yangtze River Pharmaceutical, Taizhou, China) every other day by gavage, and the other group was administered an equal volume of PBS as a control. Tumor size was measured every 4 d. After 18 d of tamoxifen treatment, the mice were euthanized, and the tumor tissue was stripped and fixed in 4% paraformaldehyde for further hematoxylin-eosin staining and immunohistochemistry analyses. Briefly, paraffin-embedded sections were placed in an antigen retrieval solution and allowed to stand in 3% hydrogen peroxide. The tissue was blocked at 37 °C with 5% BSA for 1 h, and the primary antibodies targeting TRAIL, C-CAS3, and Ki67 were blocked on the tissue sections overnight as instructed. The next day, the sections were incubated with the secondary antibodies for 1 h at 37 °C, then the sections were stained with diaminobenzidine and counterstained with hematoxylin. Finally, the sections were dried, placed under a light microscope, and images were taken. All antibodies are listed in Appendix A.

### 2.9. Colony-Formation Assay

Here, 1000 cells were seeded into 6 cm plates and treated with 20 μM 4-OHT or PBS. The medium was changed every 3 d. After 14 d of incubation, colonies were fixed with 4% paraformaldehyde for 15 min and stained with 0.1% crystal violet (C0775-25G, Sigma) at room temperature. ImageJ software v1.8 (National Institutes of Health, Bethesda, MD, USA) was used to count the number of colonies.

### 2.10. Flow Cytometry Analysis of Cell Apoptosis

Cell apoptosis was analyzed using the annexin V-fluorescein isothiocyanate/propidium iodide apoptosis detection kit (556547, BD Biosciences) according to the manufacturer’s instructions. MCF7 and T47D cells were treated with 10 μM TAM or ethanol as a control for 48 h, stained with annexin V-fluorescein isothiocyanate and propidium iodide, and analyzed by a fluorescence-activated cell sorting using a FACS Calibur (BD Biosciences). The cell apoptosis data were analyzed using FlowJo V10.7.1 software (Tree Star, San Francisco, CA, USA).

### 2.11. Cell Proliferation Assay

For the cell counting Kit-8 (CCK-8) assay, 4000 per well cells were seeded in a 96-well plate. Cells were treated with different concentrations of 4-0HT for 48 h (each concentration in four replicate wells). After 48 h of incubation, the CCK-8 reagent (40203ES80, Yeasen Biotechnology, Shanghai, China) was used to detect the OD value of each well at 450 nm absorbance. The 5-ethynyl-20-deoxyuridine (EdU) assay kit (E-CK-A218, Beyotime Biotechnology, Shanghai, China) was used to detect cell proliferation. Cells were incubated with an EdU buffer and stained with 4′,6-diamidino-2-phenylindole following the manufacturer’s protocol.

### 2.12. Gene Ontology (GO) and Kyoto Encyclopedia of Genes and Genomes (KEGG) Enrichment Analyses of Target Genes

GO and KEGG enrichment analyses were performed to determine the functions and biological pathways of the predicted target genes. The GO terms and KEGG pathways for the tRF-16-K8J7K1B target genes were retrieved from http://www.geneontology.org (accessed on 10 February 2021) and http://www.genome.jp/kegg/ (accessed on 10 February 2021), respectively. The results with corrected *p* < 0.05 were considered significant.

### 2.13. Statistical Analysis

Here, SPSS Statistics 26.0 (IBM SPSS, Chicago, IL, USA) and GraphPad prism V8.0.2 (GraphPad Software, San Diego, CA, USA) software were used for statistical analyses. Survival curves were established using K–M analysis and compared using a two-tailed log-rank test. Univariate and multivariate Cox proportional-hazards regression models were performed using IBM SPSS Statistics 26. All experimental data are presented as (mean ± standard deviation). Differences between groups were compared using Student’s *t*-test or one-way analysis of variance. * *p* < 0.05, ** *p* < 0.01, *** *p* < 0.001, and **** *p* < 0.0001 were considered statistically significant.

## 3. Results

### 3.1. tRF-16-K8J7K1B Is Upregulated in the Breast Cancer Cells and Serums of Patients with Tamoxifen-Resistant Breast Cancer

To explore the correlation between tRFs, tiRNAs, and tamoxifen resistance in breast cancer, three pairs of tamoxifen-sensitive and tamoxifen-resistant breast cancer cell lines were used for high-throughput sequencing to screen for differentially expressed tRFs and tiRNAs. According to the sequence data, 158 tRFs and tiRNAs were identified with a fold change greater than 2.0, *p* < 0.05 (Figure 1A, Appendix A). Combined with the existing MINTbase v2.0 (https://cm.jefferson.edu/MINTbase, accessed on 18 September 2019) database, tsRBase (http://www.tsrbase.org, accessed on 18 September 2019), and tRFdb (http://genome.bioch.virginia.edu/trfdb/search.php, accessed on 18 September 2019), after removing pieces with incomplete annotation information or low expression abundance, we screened 10 differentially expressed tRFs and tiRNAs. Among them, seven were upregulated in tamoxifen-resistant cell lines (tRF-16-K8J7K1B, tRF-16-489B3RB, tRF-17-WSNKP92, tRF-18-HRE9XFD2, tRF-18-BS68BFD2, tRF-18-HR0VX6D2, and tRF-28-6978WPRLXND5 tRF-16-K8J7K1B), and three were downregulated (tRF-32-P4R8YP9LON4V3, tRF-32-PNR8YP9LON4V3, and tRF-30-R9JP9P9NH5HY). Subsequently, we used the two tamoxifen-resistant cell lines, MCF-7-TR and T47D-TR, to verify the expression of the candidate tRFs and tiRNAs. Reverse transcription–quantitative polymerase chain reaction (RT-qPCR) indicated that, in contrast to the sensitive cells, the expression of tRF-16-K8J7K1B was upregulated in both tamoxifen-resistant cells with the most significant difference (Figure 1B,C). Therefore, we selected tRF-16-K8J7K1B for further research.

To explore the clinical application value of tRF-16-K8J7K1B, we detected the expression of tRF-16-K8J7K1B in serum samples from 56 HR+ early patients with breast cancer treated with tamoxifen (27 tamoxifen-sensitive patients and 29 tamoxifen-resistant patients). Compared with tamoxifen-sensitive patients, tRF-16-K8J7K1B was significantly upregulated in the serum of tamoxifen-resistant individuals (Figure 1D). Additionally, we discovered that a higher expression of tRF-16-K8J7K1B was correlated with worse disease-free survival (DFS) in tamoxifen-treated patients (median value used as the cutoff value) (Figure 1E). Figure 1F reveals that the receiver operating characteristic curve for tRF-16-K8J7K1B had an area under the curve of 0.7733 (*p* = 0.00004, 95% confidence interval = 0.6531–0.8936), suggesting the great potential of tRF-16-K8J7K1B as a noninvasive circulating biomarker for tamoxifen resistance. Univariate analyses identified tRF-16-K8J7K1B as a predictive factor associated with tamoxifen response in HR+ breast cancer (*p* = 0.0033). Multivariate analyses further revealed that a higher expression of tRF-16-K8J7K1B and Ki67 index were associated with poor prognosis in tamoxifen-treated patients (Table 1). Based on these results, tRF-16-K8J7K1B may be an independent prognostic factor and a therapeutic target for tamoxifen-resistant breast cancer.

### 3.2. tRF-16-K8J7K1B Promotes Tamoxifen Resistance in Breast Cancer

To investigate the role of tRF-16-K8J7K1B in tamoxifen resistance, we overexpressed tRF-16-K8J7K1B in MCF7 and T47D cells and knocked down tRF-16-K8J7K1B in resistant cells (Figure 2A and Appendix A). Transiently transfected cells were used to detect the effect of tRF-16-K8J7K1B on cell proliferation and apoptosis following tamoxifen treatment. Subsequently, after 48 h of treatment with 20 μM 4-OHT, the overexpression of tRF-16-K8J7K1B increased the half-maximal inhibitory concentration (IC50) of tamoxifen in tamoxifen-sensitive cells, and tamoxifen-resistant cells with tRF-16-K8J7K1B had lower IC50 values than the control group (Figure 2B and Appendix A). Moreover, clone formation, EdU, and flow cytometry assays revealed that an overexpression of tRF-16-K8J7K1B resulted in increased cell proliferation and a lower rate of drug-induced cell apoptosis in tamoxifen-sensitive cells, whereas a knockdown of tRF-16-K8J7K1B resulted in decreased proliferation ability and a higher rate of drug-induced cell apoptosis in tamoxifen-resistant cells (Figure 2C–E and Appendix A). These results indicate that tRF-16-K8J7K1B could promote tamoxifen resistance in breast cancer cells.

### 3.3. tRF-16-K8J7K1B Can Be Secreted by Tamoxifen-Resistant Cells and Transmitted to Sensitive Cells via Exosomes

Exosomes are small vesicles that are secreted by various cell types. Some studies have demonstrated that ncRNAs can be transferred from cell to cell via exosomes, affecting the biological behavior of recipient cells [21]. This study investigated whether tRF-16-K8J7K1B promotes tamoxifen resistance via its incorporation into exosomes. First, we isolated and identified exosomes derived from tamoxifen-resistant cells. Transmission electron microscopy and Western blotting revealed that the exosomes had a typical cup-shaped morphology (Figure 3A and Appendix A), with an enriched expression of exosome marker proteins, namely ALIX, TSG101, CD9, and CD63 (Figure 3B). Next, we labeled exosomes derived from tamoxifen-resistant cells with PKH67 and co-cultured them with sensitive cells for more than 48 h. Laser confocal microscopy confirmed that the recipient cells internalized the labeled exosomes (Figure 3C and Appendix A). Subsequently, we compared the relative expression of tRF-16-K8J7K1B in the exosomes. RT-qPCR indicated that tRF-16-K8J7K1B was markedly higher in TR-exo (exosomes derived from tamoxifen-resistant cells) than in S-exo (exosomes derived from tamoxifen-sensitive cells) (Figure 3D). After co-culturing sensitive MCF7 and T47D cells with exosomes derived from different donor cells, we verified the elevated expression of tRF-16-K8J7K1B in tamoxifen-sensitive cells caused by TR-exo. However, this effect was eliminated when incubated with exosomes from tRF-16-K8J7K1B-knockdown tamoxifen-resistant cells (Figure 3E and Appendix A). Moreover, our study demonstrated that tamoxifen-sensitive MCF-7 cells exhibited decreased sensitivity to tamoxifen treatment after co-culturing with TR-exo. This effect was reversed after culturing with exosomes derived from knocked-down tRF-16-K8J7K1B in tamoxifen-resistant cells (Figure 3F). These results were also verified in the T47D cell line (Appendix A), suggesting that exosomal tRF-16-K8J7K1B derived from tamoxifen-resistant cells can promote tamoxifen resistance in recipient cells.

### 3.4. TRAIL Is Responsible for tRF-16-K8J7K1B-Mediated Tamoxifen Resistance

To further explore the possible mechanism by which tRF-16-K8J7K1B promotes tamoxifen resistance, we used RNA sequencing to identify the target genes of tRF-16-K8J7K1B. The heatmap revealed the top 20 differentially expressed genes in tRF-16-K8J7K1B-overexpressed MCF7 cells compared to the control group (Figure 4A). Combined with GO and KEGG enrichment analysis, we identified TRAIL as the target of tRF-16-K8J7K1B (Figure 4B,C). The sequencing results were verified using Western blotting and RT-qPCR (Figure 4D,E). The ACTD experiment also proved that tRF-16-K8J7K1B decreased the expression of TRAIL over time (Figure 4F). Next, we searched for a possible binding site between tRF-16-K8J7K1B and TRAIL using website prediction (https://bibiserv.cebitec.uni-biele.org.de/rnahybrid, accessed on 5 November 2022) (Figure 4G). The luciferase receptor assay revealed that the tRF-16-K8J7K1B mimic attenuated the luciferase activity of 293T cells co-transfected with the wild-type plasmid but had no effect on the mutant plasmid (Figure 4H). These results demonstrate that tRF-16-K8J7K1B inhibited the expression of TRAIL by directly binding to its 3′-UTR region.

### 3.5. Exosomal tRF-16-K8J7K1B Promotes Tamoxifen Resistance in Breast Cancer by Targeting TRAIL

To date, experiments have demonstrated the regulatory effect of tRF-16-K8J7K1B on TRAIL. Next, we investigated the role of TRAIL in tamoxifen resistance. We used three different small interfering RNAs (siRNAs) to knock down TRAIL in MCF7 and T47D cells and selected si-217, which had better knockdown efficiency, for the subsequent experiments (Figure 5A,B and Appendix A). Consistent with previous results, TRAIL siRNA-transfected cells had a decreased sensitivity to tamoxifen treatment compared to the control group (Figure 5C–F and Appendix A). Moreover, co-transfection with TRAIL siRNA partially reversed the effects of the tRF-16-K8J7K1B inhibitor on cell proliferation and apoptosis (Figure 5G,H and Appendix A). To further explore whether exosomal tRF-16-K8J7K1B promotes tamoxifen resistance in breast cancer by downregulating TRAIL expression, we extracted exosomes derived from MCF7/S, MCF7/TR, MCF7/TR-NC, and MCF7/TR-IN cells and performed co-culture experiments. After more than 48 h, cells treated with TR-exo had lower expression of TRAIL, cleaved caspase 3 and cleaved poly (ADP-ribose) polymerase (PARP), whereas IN-exo did not downregulate these apoptosis-related proteins (Figure 5I and Appendix A). Additionally, we discovered that the tRF-16-K8J7K1B inhibitor upregulated the protein expression of TRAIL, cleaved caspase 3, and cleaved PARP in MCF7 and T47D cells compared to their control group, and these effects were partially reversed after co-transfection with TRAIL siRNA (Figure 5J and Appendix A). These results fully endorse our point that exosomal tRF-16-K8J7K1B downregulates the expression of TRAIL to promote tamoxifen resistance, which is achieved by reducing drug-induced cell apoptosis.

### 3.6. Exosomal tRF-16-K8J7K1B Promotes Tamoxifen Resistance In Vivo

To further explore the function of exosomal tRF-16-K8J7K1B in vivo, we established breast cancer in suit models with BALB/c nude mice. When the tumor volume reached 50 mm^3^, mice were injected intratumorally with PBS, S-exo, TR/NC-exo, or TR/IN-exo (Figure 6A). Figure 6B–D revealed that the knockdown of tRF-16-K8J7K1B significantly decreased tumor weight and volume compared to the NC-exo-treated group after treatment with the same dose of tamoxifen. The above results indicate that exosomes derived from drug-resistant cells could weaken the inhibitory effect of tamoxifen on tumor growth. The results of HE staining and immunohistochemistry revealed that the expressions of TRAIL, cleaved caspase 3, and Ki67 were lower in the tumor tissues of the mice treated with NC-exo. More importantly, after 18 d of TAM treatment, the expression of these proteins was not significantly upregulated in the NC-exo-treated group (Figure 6E–G). These results further prove that exosomes derived from TAM-resistant cells contain higher levels of tRF-16-K8J7K1B, which could decrease the sensitivity of breast cancer cells to tamoxifen by downregulating the expression of apoptosis-related proteins in the recipient cells.

## 4. Discussion

This study identified that tRF-16-K8J7K1B, a novel small ncRNA derived from the 3′-end of tRNA^Ala-TGC^, was highly expressed in tamoxifen-resistant cells compared to parental cells using high-throughput sequencing data. High tRF-16-K8J7K1B expression was associated with shorter DFS in HR+ patients treated with tamoxifen. Gain- and loss-of-function assays indicated that tRF-16-K8J7K1B promotes tamoxifen resistance in breast cancer cells. Mechanistically, extracellular tRF-16-K8J7K1B confers tamoxifen resistance via incorporation into exosomes, and then degrades the expression of apoptosis-related proteins by directly binding to the 3′UTR region of TRAIL, reducing the proportion of drug-induced cell apoptosis, thereby inducing tamoxifen resistance (Figure 7).

Tamoxifen resistance has brought challenges to treating patients with HR+ breast cancer. Although effective therapies, such as aromatase inhibitor and CDK4/6 inhibitors, are now available, tamoxifen resistance continues to occur; moreover, there are currently no definitive biomarkers available to predict response to tamoxifen. Thus, explorations of the molecular mechanisms underlying tamoxifen resistance is of great scientific and therapeutic importance.

Functionally, the dysregulation of ncRNAs is associated with aberrant biological behavior in human cancers. The role of ncRNAs in tamoxifen resistance has been extensively studied [22,23]. Recently, tRFs have attracted the attention of scientists as a new type of small ncRNAs. Studies have revealed that the aberrant expression of tRFs may be related to dysregulated biological processes in tumor cells, such as proliferation, invasion, apoptosis, and self-renewal [24]. Some newly identified tRFs exist in human serum and have been considered new biomarkers and therapeutic targets for treating human cancer. These findings expand our knowledge of the role of tRFs in cellular pathophysiological processes. However, to date, the relationship between tRFs and tamoxifen resistance in breast cancer remains unclear.

In this study, we discovered that tRF-16-K8J7K1B was upregulated in tamoxifen-resistant cells and patient sera. This result suggests that tRF-16-K8J7K1B plays a crucial role in promoting tamoxifen resistance. The overexpression of tRF-16-K8J7K1B effectively promoted cell proliferation, inhibited apoptosis, and reduced the sensitivity of cells to tamoxifen. Additionally, our study discovered that tRF-16-K8J7K1B was relatively stable and convenient for detection in human serum. Thus, we believe that tRF-16-K8J7K1B has the potential to be a predictive biomarker and therapeutic target for tamoxifen resistance in breast cancer and is worthy of further exploration.

Exosomes are small molecular vesicles that mediate cellular information exchange. Interestingly, some components of drug-resistant cells, such as miRNAs, can be transmitted by exosomes and promote drug resistance in recipient cells [25]. Previous studies have demonstrated that exosomal lncARSR promotes sunitinib-resistant renal cell carcinoma by upregulating the expression of AXL and c-MET in recipient cells [15]. Exosomal AFAP1-AS1 confers the trastuzumab-resistant phenotype in breast cancer cells by inducing the translation of the ERBB2 protein [26]. However, it has not been demonstrated whether tRFs can be transmitted between cancer cells via exosomes. Here, we discovered that tRF-16-K8J7K1B derived from tamoxifen-resistant cells can be incorporated into exosomes and transmitted to sensitive cells. We verified that exosomal tRF-16-K8J7K1B could induce tamoxifen resistance in receptor cells, demonstrating a novel mechanism by which tRF spreads tamoxifen resistance.

The functional modes of tRNA derivatives are multiplex and have not been fully elucidated [27]. Previous studies have suggested that tRFs could affect the expression of target genes by forming an RNA-induced silencing complex (RISC) or influence protein translation by acting on translation initiation factors [28,29,30]. Among them, targeting the 3′-UTR region of the target gene is the most widely studied mechanism of tRFs in regulating the occurrence, development, and drug resistance of human cancers [31]. Therefore, we used RNA sequencing to identify the target genes of tRF-16-K8J7K1B and identified TRAIL as a possible downstream target of tRF-16-K8J7K1B. Combined with bioinformatics analyses and luciferase assays, we proposed that tRF-16-K8J7K1B promotes tamoxifen resistance in breast cancer by inhibiting drug-induced cell apoptosis. Further experiments have demonstrated that tRF-16-K8J7K1B could downregulate TRAIL at the mRNA and protein levels by directly binding to its 3′UTR region, thereby promoting tamoxifen resistance.

TRAIL is a member of the tumor necrosis factor superfamily, and initiates apoptosis by triggering the activity of pro-apoptotic proteins, such as caspase 3 and PARP [19]. Importantly, the induction of apoptosis by TRAIL mainly occurs in transformed and tumor cells, but it is ineffective in most normal cells [32]. Therefore, TRAIL has received widespread attention as a core component of cancer treatment [33]. Previous studies have revealed that stronger cell proliferation ability and lower drug-induced cell apoptosis are correlated with tamoxifen resistance [34], and that upregulating the expression could inhibit the progression of breast cancer and hepatocellular carcinoma [35]. Here, we used a series of gain- and loss-of-function experiments to evaluate whether TRAIL is related to tamoxifen resistance in breast cancer. Our data indicated that inhibiting tRF-16-K8J7K1B led to a higher apoptosis rate and decreased sensitivity to tamoxifen in breast cancer cells. Notably, knocking down TRAIL partially reversed these effects. These results support our hypothesis that exosomal tRF-16-K8J7K1B is a potential biomarker and therapeutic agent for tamoxifen resistance.

We believe that this study provides advances in this field; however, our work has some limitations. First, we did not explore the mechanism by which tRF-16-K8J7K1B is increased in tamoxifen-resistant cells, and whether tRF-16-K8J7K1B has the same role in other types of cancer remains unclear. Second, a regimen to efficiently target tRF-16-K8J7K1B in breast cancer is yet to be defined. Whether tRF-16-K8J7K1B plays a similar role in breast cancer treated with aromatase inhibitors or CDK4/6 inhibitors remains unknown and requires further investigation.

## 5. Conclusions

In conclusion, our study illustrates that tRF-16-K8J7K1B confers tamoxifen resistance in breast cancer cells via packaging into exosomes. Mechanistically, exosomal tRF-16-K8J7K1B downregulated the expression of apoptosis-related proteins by targeting TRAIL, resulting in decreased cell apoptosis and tamoxifen resistance. Breast cancer patients with a high expression of tRF-16-K8J7K1B have poor prognosis. Our research identified exosomal tRF-16-K8J7K1B as a potential predictive biomarker and therapeutic target to overcome tamoxifen resistance, which may further improve the outcomes of patients with ER-positive breast cancer.

## Figures and Tables

**Figure 1 cancers-15-00899-f001:**
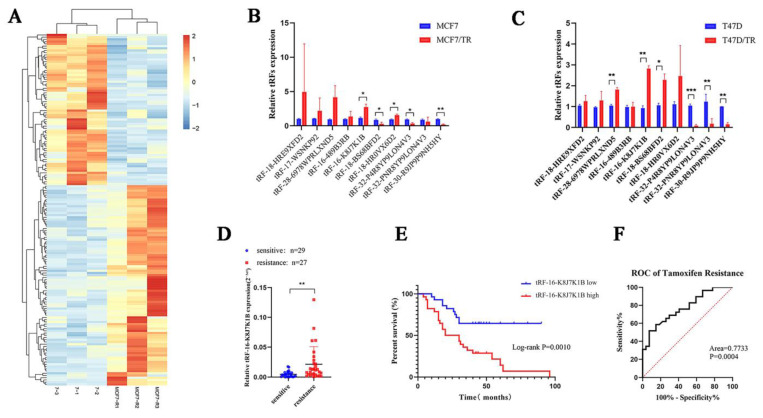
tRF-16-K8J7K1B is upregulated in tamoxifen-resistant breast cancer patients and correlates with shorter DFS in patients. (**A**) Heatmaps of differently expressed tRFs and tiRNAs in tamoxifen-sensitive and resistant cell lines. (**B**) The 10 candidate differentially expressed tRFs and tiRNAs in MCF7/S and MCF7/TR cells. (**C**) The 10 candidate differentially expressed tRFs and tiRNAs in T47D/S and T47D/TR cells. (**D**) RT-qPCR analysis revealed the relative expression of tRF-16-K8J7K1B in serum of tamoxifen-sensitive and tamoxifen-resistant patients. (**E**) Patients with high levels of tRF-16-K8J7K1B expression had significantly worse DFS. (**F**) ROC curve analysis showed the power of tRF-16-K8J7K1B in predicting tamoxifen resistance. Data are presented as mean ± SEM. * *p* < 0.05; ** *p* < 0.01; *** *p* < 0.001.

**Figure 2 cancers-15-00899-f002:**
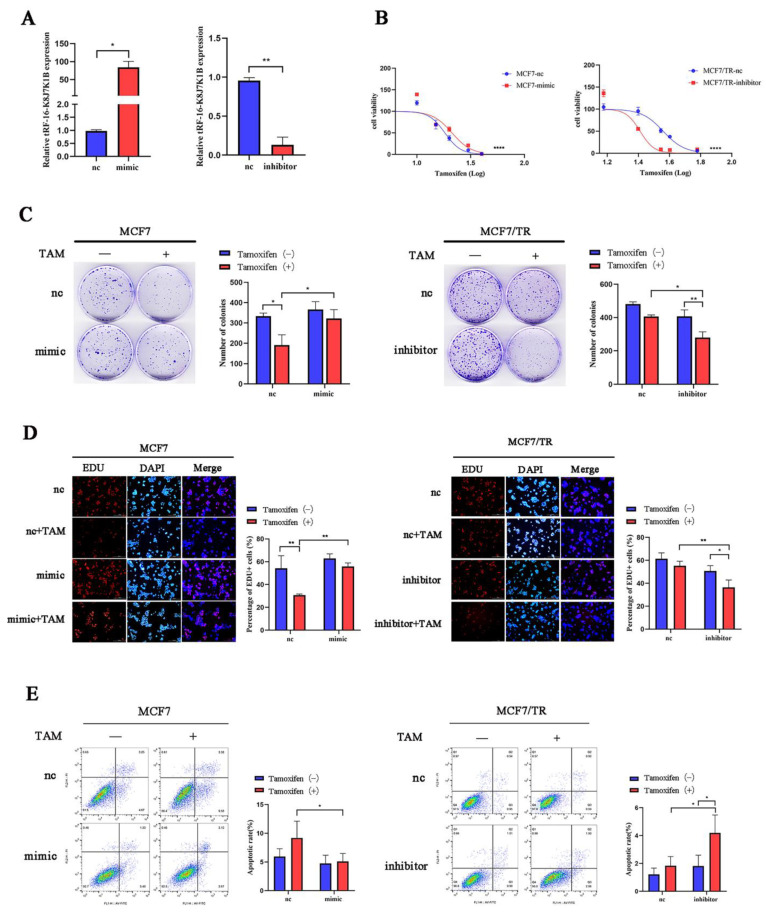
tRF-16-K8J7K1B promotes the tamoxifen resistance of breast cancer cells in vitro. (**A**) tRF-16-K8J7K1B was overexpressed or silenced by the transfection of respective vectors. RT-qPCR analyses revealed the relative expression of tRF-16-K8J7K1B and U6 in transfected cells. (**B**,**C**) The effects of tRF-16-K8J7K1B overexpression or knock down on the proliferation of MCF7 cells were examined by CCK-8 assay (**B**) and colony-formation assay (**C**). (**D**) EdU assays were used to detect the proliferation rates of MCF-7 cells after tRF-16-K8J7K1B was overexpressed or knocked down (**E**) Cells were treated with 4-OHT or ethanol (control), and flow cytometry was used to detect cell apoptosis in MCF7 and MCF7/TR cells. TAM: Tamoxifen; nc: negative control; mimic: tRF-16-K8J7K1B mimic; inhibitor: tRF-16-K8J7K1B inhibitor. Data are presented as mean ± SEM. * *p* < 0.05; ** *p* < 0.01; **** *p* < 0.0001. The scale bar represents 50 μm.

**Figure 3 cancers-15-00899-f003:**
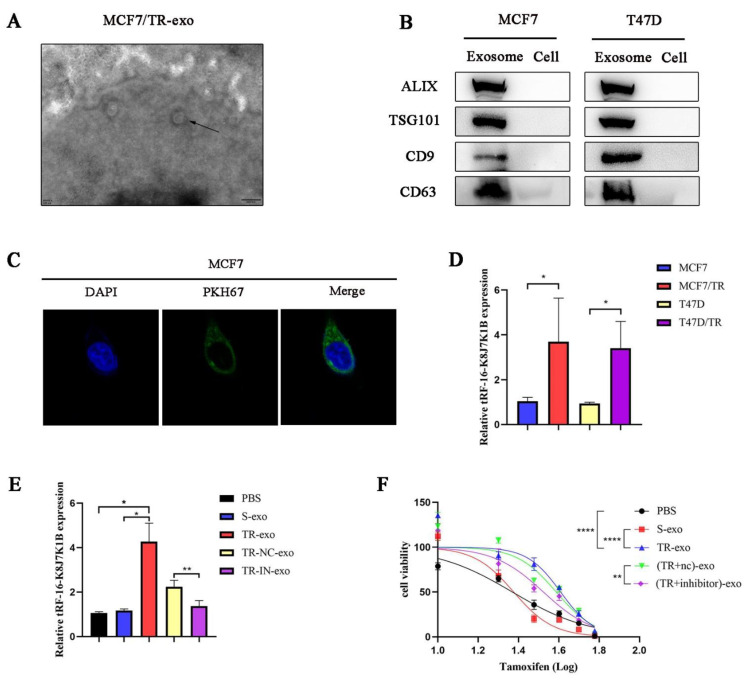
Exosomes mediate the delivery of tRF-16-K8J7K1B and promote tamoxifen resistance. (**A**) The morphology of exosomes derived from MCF7/TR cells under transmission electron microscope. The arrow indicated the exosomes. Scale bar, 200 nm. (**B**) Western blotting revealed the expression of exosome marker proteins ALIX, TSG101, CD9, and CD63 in suspension extracted by ultracentrifugation. (**C**) Laser confocal microscope revealed the uptake of PKH67-labeled exosomes (green fluorescence) derived from MCF7/TR cells by recipient cells. Cell nucleus was labeled with DAPI (blue fluorescence). (**D**) RT-qPCR analysis revealed the expression of tRF-16-K8J7K1B in exosomes derived from tamoxifen-sensitive cells and their respective resistant cells. (**E**) RT-qPCR detected the expression of tRF-16-K8J7K1B in recipient MCF7 cells that were treated with PBS, S-exo, TR-exo, NC-exo (TR-exo with tRF-16-K8J7K1B-nc), and IN-exo (TR-exo with tRF-16-K8J7K1B-inhibitor). (**F**) CCK-8 assay was performed to analyze tamoxifen sensitivity of MCF7/S cells after co-culturing with exosomes according to the indicated factors. Data are presented as mean ± SEM. * *p* < 0.05; ** *p* < 0.01; **** *p* < 0.0001. The uncropped blots are shown in Appendix A.

**Figure 4 cancers-15-00899-f004:**
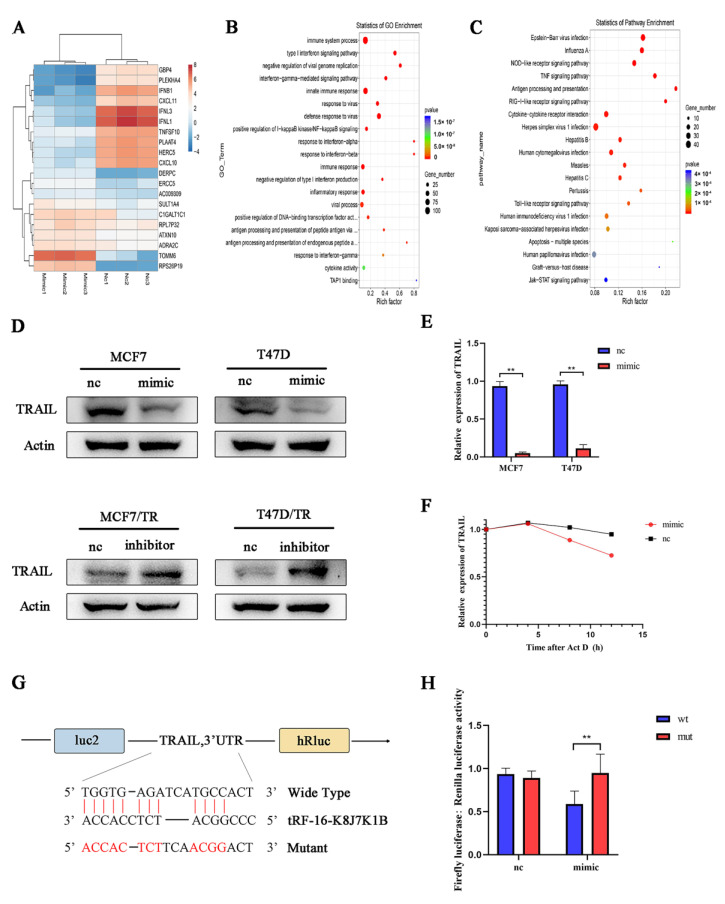
TRAIL is the target gene of tRF-16-K8J7K1B. (**A**) Heat map shows the change in mRNA expression in tRF-16-K8J7K1B-overexpressed MCF7 cells compared with the control group. (**B**) GO analysis and (**C**) KEGG analysis indicate the enriched pathways by the target genes of tRF-16-K8J7K1B. (**D**) Western blotting and (**E**) RT-qPCR revealed the expression of TRAIL in MCF7 and T47D cells transfected with tRF-16-K8J7K1B mimic or control. (**F**) RT-qPCR detected the expression of TRAIL in MCF7 cells after 0, 4, 8, and 12 h of ACTD treatment. (**G**) The predicted tRF-16-K8J7K1B binding site in TRAIL. (**H**) Luciferase reporter assay was performed to detect the luciferase activity of WT or MUT TRAIL 3′ UTR after co-transfection with tRF-16-K8J7K1B mimics or negative control. Data are presented as mean ± SEM. ** *p* < 0.01. The uncropped blots are shown in Appendix A.

**Figure 5 cancers-15-00899-f005:**
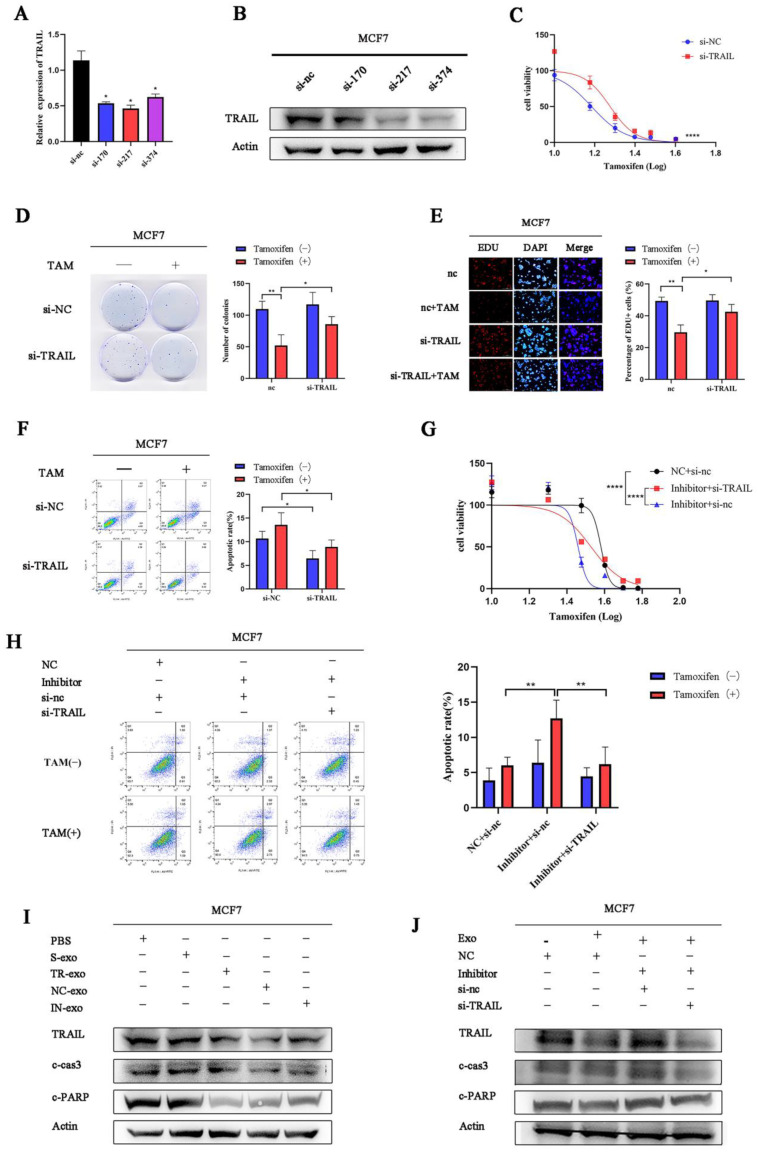
Exosomal tRF-16-K8J7K1B promotes tamoxifen resistance by targeting TRAIL in vitro. (**A**,**B**) RT-qPCR and Western blotting detected the knockdown effect on TRAIL by three different siRNAs in MCF7 cells. (**C**–**E**) The effects of TRAIL knockdown on the proliferation of MCF7 cells were examined by CCK-8 assay (**C**), colony-formation assays (**D**), and EdU assays (**E**). (**F**) Flow cytometry revealed cell apoptosis rate in MCF7 cells transfected with TRAIL’s siRNA. (**G**,**H**) CCK-8 assay and flow cytometry were used to detect tamoxifen sensitivity of MCF7 cells transiently transfected with TRAIL siRNA, combined with tRF-16-K8J7K1B inhibitor or tRF-16-K8J7K1B. (**I**) Western blotting revealed the expression of TRAIL, cleaved caspase 3, and cleaved PARP after co-culturing with exosomes in MCF7 cells. (**J**) Co-transfection with TRAIL’s siRNA decreased the expression of indicated apoptosis-related protein after treatment with TR/IN-exo in MCF7 cells. Data are presented as mean ± SEM. * *p* < 0.05; ** *p* < 0.01; **** *p* < 0.001. The uncropped blots are shown in Appendix A.

**Figure 6 cancers-15-00899-f006:**
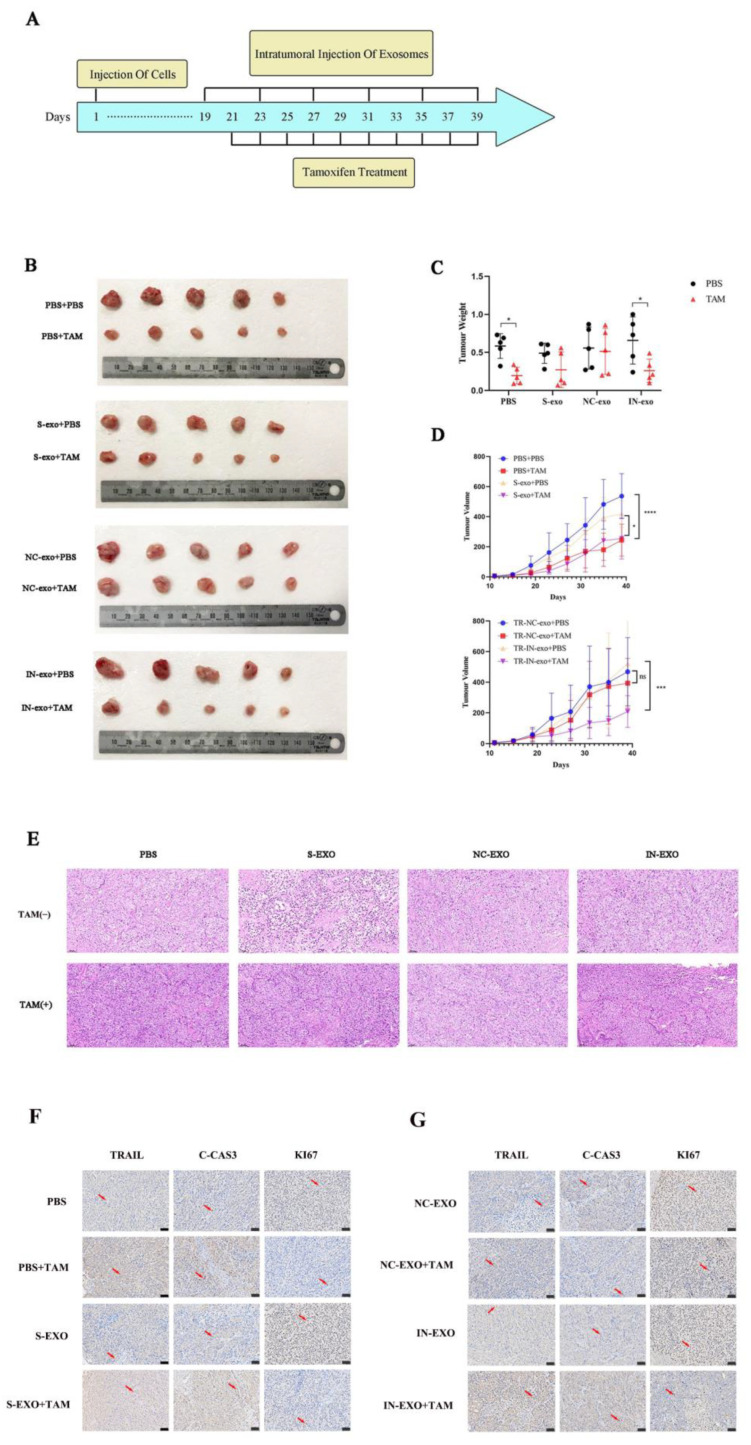
Exosomal tRF-16-K8J7K1B promote tamoxifen resistance in vivo. (**A**) Schematic diagram of animal experiment. (**B**) After exosomes and tamoxifen treatment, tumors isolated from mice are shown in images. (**C**) Tumor weight at the endpoint. (**D**) Tumor volume changes during our study. (**E**) Tumor tissue slices were stained with hematoxylin-eosin. (**F**,**G**) Immunohistochemistry analyses of the expression of TRAIL, cleaved caspase 3, and Ki67 in tumor tissues. The red arrows indicate the positive immunohistochemical reactions. S-exo: exosomes derived from tamoxifen-sensitive cells; NC-exo: exosomes derived from tamoxifen-resistant cells transfected with tRF-16-K8J7K1B NC; IN-exo: exosomes derived from tamoxifen-resistant cells transfected with tRF-16-K8J7K1B inhibitor; C-CAS3: cleaved caspase 3. Scale bars: 50 μm. Data are presented as mean ± SEM. * *p* < 0.05; *** *p* < 0.001; **** *p* < 0.001; ns: no significance.

**Figure 7 cancers-15-00899-f007:**
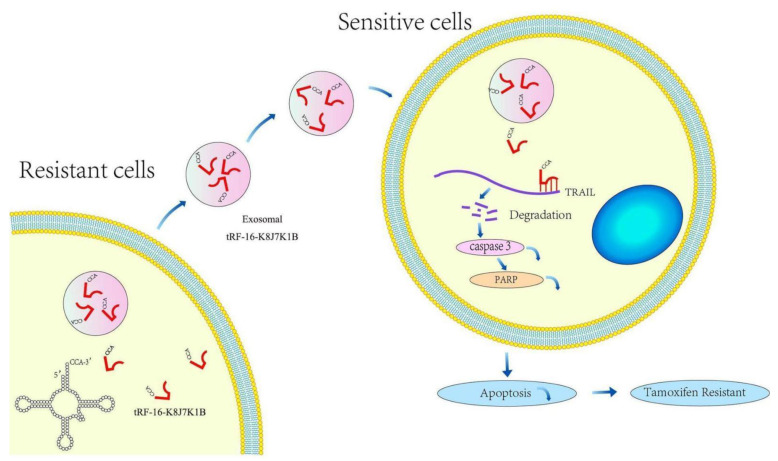
Schematic diagram of the proposed mechanisms. tRF-16-K8J7K1B was upregulated in tamoxifen-resistant BC cells and could be taken up by tamoxifen-sensitive cells via encapsulation into exosomes. Exosomal tRF-16-K8J7K1B promotes tamoxifen resistance by degrading TRAIL mRNA, leading to decreased expression of pro-apoptotic proteins caspase 3 and PARP in recipient cells.

**Table 1 cancers-15-00899-t001:** Univariate and multivariate analyses of the clinicopathological factors for PFS in HR+ breast cancer. * *p* < 0.05; *** *p* < 0.001.

Risk Factors	Univariate Analyses *p*-Value	Multivariate Analyses		
		HR	*p*-Value	95% CI
tRF-16-K8J7K1B (ΔCT ≤ 7.5, >7.5)	0.0001 ***	2.565	0.02 *	1.157–5.685
Ki-67 score (≤30%, >30%)	0.1707	2.687	0.037 *	1.059–6.815
Histologic grade (I, II, III)	0.265	2.013	0.081	0.918–4.412
T stage (T1/T2, T3/T4)	0.3516			
Lymph node metastasis (No, Yes)	0.7458			
Age (≤50, >50)	>0.9999			

## Data Availability

All data analyzed during this study are included in this published article and its Appendix A.

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
