# Peer review of "Exosome-Transmitted tRF-16-K8J7K1B Promotes Tamoxifen Resistance by Reducing Drug-Induced Cell Apoptosis in Breast Cancer"

_cancers, 2023, doi:10.3390/cancers15030899_

Round 1

Reviewer 1 Report

* The authors investigated the important role of tRF-16-K8J7K1B in the tamoxifen resistance but an extensive English revision by a native speaker or a specialized company is required. I have some issues regarding your manuscript: 

* The title of the manuscript should be clear and not contain any abbreviation to attract readers.

 * Line 25: the abbreviation (( tRF-16-K8J7K1B)) must be defined in its first mention. Please, also revise all abbreviations in the abstract and the main manuscript body. 

* Line 31: there is no other used (TRAIL) in the abstract so it must not write as an abbreviation.

* The authors should introduce tumor necrosis factor-related apoptosis-inducing ligand in their introduction and previous studies.

* Line 93: the authors should mention the ethical approval number provided by Nanjing Medical University.

* Line 99-100: This study was approved by the Ethics 99 Committee of the First Affiliated Hospital of Nanjing Medical University. This sentence is repeated and redundant. 

* You must mention catalog numbers and module numbers for all used kits and devices.

* Line 128: the procedures for transmission electron microscopy should be mentioned.

* Please, write the function of each antibody in Table S3.

* Line 161: Please, provide the ethical approval number.

* Line 165: Please, define MCF7/TR-nc in its first mention.

* Line 171: Please, mention the used antibodies and the used immunohistochemical procedures.

* Line 172: Please, measure the histopathological lesion score in hematoxylin-eosin staining and the immunohistochemical area percentage in the immunohistochemical reactions.

* Figure 2: please, define all abbreviations in the figure legends such as TAM, nc, and mimic.

* Figure 6: (E, F, G) please, put the scale bar on all figures and use the arrows to indicate the histopathological changes between the different groups and define all the used abbreviations. 

Reviewer 2 Report

The manuscript submitted by Sun C, et al. is a description of a molecular mechanism that promotes drug resistance in breast cancer. By using a combination of biochemical and molecular in vitro techniques, in vivo mouse models of breast cancer, and statistical analyses the authors found a novel mechanism that confers resistance to tamoxifen. The mechanism is mediated by transfer of RNA fragments, contained within exosomes, that down-modulate the expression of apoptosis-related proteins. This manuscript contributes to recent observations on the exosome mediated transfer of RNA species (micro RNA, long non coding RNA, and circular RNA), as the basis of mechanisms that confer drug resistance in breast cancer. The topic is relevant as recent efforts are focused in identifying exosomal non-coding RNAs and their potential targets that modulate tumour features such as metastatic potential and resistance to drugs. The significance and potential implications of understanding the mechanism that confer drug resistance are very essential for improvements of patient’s clinical management.

The article is well-written, and the presentation of results well-structured and following a logical order: from identification of up-regulated RNA species that are risk factors and are associated with drug resistance, to the description of the mechanism that leads to down-modulation of the expression of pro-apoptotic proteins. The figures are well presented and with clear figure legends.

Regarding Figure 3B (showing markers of exosomes), did the authors used one marker that is exclusively present in cell fractions but not in exosomes? As it is known that several cell compartments can produce extracellular vesicles, it would be recommended to test for the presence of markers for endoplasmic reticulum and Golgi apparatus. This would guarantee that there is no contamination of the samples analysed and strengthen the statement that all RNA species are exclusively exosome derived.

With this submission there were no figure legends for the supplementary figures. It would be interesting to know what is presented in Figure S2A. What are structures surrounding what it seems to be an exosome highlighted with an arrow?

Perhaps the size of panel B and C in Figure 4 could be increased as the labelling is so small that it is difficult to read.

For Figure 5 panel J, can you improve the quality of the image for c-cas3 western blot data? The image presented is blurry and cut over the bands.

Minor comments:

Line 63: change the word “expressed”. tRFs are not expressed in serum. They might be present or increased. The use of the word “expressed” in this context is not correct.

Line 109: Change GIBICO for GIBCO

In Conclusions section line 490, there are some words missing. The sentence in lines 489-491 is dificult to understand.

Round 2

Reviewer 1 Report

The authors made extensive improvement in their manuscript. Another quick language revision is required. 

Reviewer 2 Report

No further comments.